# Rethinking Thinking Tokens: LLMs as Improvement Operators

**Lovish Madaan** [1 2 *]   **Aniket Didolkar** [3]   **Suchin Gururangan** [4]   **John Quan** [5]   **Ruan Silva** [5]   **Ruslan Salakhutdinov** [6]
**Manzil Zaheer**   **Sanjeev Arora** [7]   **Anirudh Goyal** [5]

## Abstract

Reasoning training incentivizes LLMs to produce long chains of thought (long CoT), which among other things, allows them to explore solution strategies with self-checking. This results in higher accuracy, but inflates context length, token/compute cost, and answer latency. We ask: *Can current models leverage their metacognition to provide other combinations on this Pareto frontier, e.g., better accuracy with lower context length and/or latency?* Abstractly, we view the model as an *improvement operator* on its own "thoughts" with a continuum of possible strategies. We study an inference family **Parallel-Distill-Refine (PDR)**, which performs the following: (i) generate diverse drafts in parallel; (ii) *distill* them into a bounded, textual workspace; and (iii) *refine* conditioned on this workspace, producing an output that seeds the next round. Importantly, context length (hence compute cost) is controllable via degree of parallelism, and is no longer conflated with the total number of generated tokens. We report **PDR** instantiations of current models that give better accuracy than long CoT while incurring lower latency. Setting degree of parallelism to 1 yields a subcase **Sequential Refinement (SR)** (iteratively improve a single candidate answer) which provides performance superior to long CoT (at the cost of higher latency). Success of such model orchestrations raises the question whether further training could shift the Pareto frontier. To this end, we train an $8B$ thinking model with Reinforcement Learning (RL) to make it consistent with **PDR** as the inference method. On math tasks with verifiable answers, iterative pipelines surpass single-pass baselines at matched sequential budgets, with **PDR** delivering the largest gains (+11% on AIME 2024 and +9% on AIME 2025).

## 1. Introduction

Scaling language models to solve harder problems has increasingly relied on eliciting explicit reasoning traces ("thinking tokens") at inference time (Wei et al., 2022; Jaech et al., 2024; Guo et al., 2025). While longer traces often correlate with accuracy, they entangle reasoning depth with sequence length and inherit long-context failure modes (Ghosal et al., 2025). In parallel, the field is gravitating towards *self-improvement*: systems that refine their own outputs via *self-directed operations* (critique, revision, debate, sample-and-select) without expert supervision (Gou et al., 2023; Du et al., 2023b; Irving et al., 2018; Yao et al., 2023; Pan et al., 2025; Zhang et al., 2025).

Stepping back from the rich body of work, one begins to see LLM inference as a malleable concept; instead of a single "reasoning trace" one encounters choices to be made from a larger pool: generate fresh answers; critique/revise/debate/summarize generated answers; create an updated answer. With this choice comes an unexplored Pareto-frontier: What is the best possible task accuracy achievable after fixing constraints on the inference process, e.g.: (i) total tokens across all generations, (ii) max depth of the generation chain ("latency"), (iii) total context length, and (iv) total compute (which depends on all of the above in complicated and system-dependent ways).

The confounding factor is that iteration alone does not guarantee progress. Simply asking the model to "try again" risks forgetting useful partial results and repeating earlier mistakes. Naïvely appending all prior attempts to the context recreates long-context failures and scales cost with the number of rounds. Current models suffer from anchoring biases (see Figure 6, 7) as well as forgetfulness. A viable scheme needs a compact state that (i) carries forward salient facts and intermediate results, (ii) flags disagreements and open subgoals, and (iii) remains bounded so each generation (and overall context-size) stays short.

We study inference strategies that generate many tokens with a compact context size. Instead of long chains of thought,

---

[*]Work done at Meta [1]Recursive [2]University College London [3]Mila, University of Montreal [4]Anthropic [5]Meta [6]Carnegie Mellon University [7]Princeton University. Correspondence to: Lovish Madaan <lvm@fastmail.com>.

*Proceedings of the $43^{rd}$ International Conference on Machine Learning*, Seoul, South Korea. PMLR 306, 2026. Copyright 2026 by the author(s).

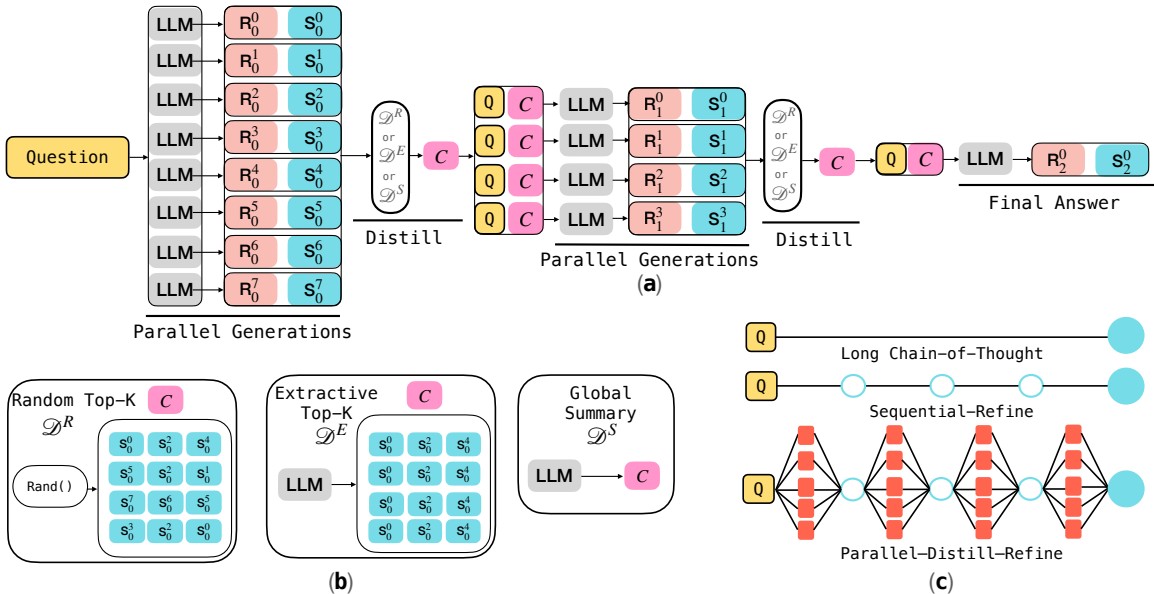

*Figure 1.* (a) **Parallel-Distill-Refine (PDR)**. In round $r$, the model generates $M_r$ parallel drafts, then distills them into a compact workspace using one of the schemes in (b); the refined state seeds the next round. (b) Distillation schemes used to build the workspace (e.g., global summary, shared top-$k$, per-sample top-$k$, random-$k$). (c) Three inference regimes. Top-**Long chain-of-thought**: a single, long trace. Middle-**Sequential Refinement (SR)**: one draft updated over short rounds. Bottom-**PDR**: each round spawns $M_r$ drafts, distills into a workspace, and refines. The example shows a 3-round configuration $M = (8, 4, 1)$ (configuration is a hyperparameter, and any other choice is possible). Across panels, the per-call *sequential budget* $B_{\text{seq}}$ (latency proxy) is held fixed, while **PDR** increases *total compute* $B_{\text{total}}$ via parallelism without increasing per-call context.

inference has phases that generate solutions within the allowed context budget and then write a bounded, round-wise summary/report (e.g., listing agreements, contradictions, intermediate results, and open subgoals). The next phase starts with only this summary and uses available workspace for fresh generations (which benefit from accumulated wisdom in the summary). Iterating this process can generate long "thinking" albeit with a bounded context size. [1]

We study two inference instantiations: (i) *Sequential refinement (SR)*, where a single artifact (solution, proof, program) is iteratively improved for a fixed number of steps; and (ii) *Parallel-Distill-Refine (PDR)* (round-wise workspace), where each round samples $M$ drafts in parallel, distills them into a bounded summary for the next round, and continues. The workspace is not persistent across rounds; it is freshly synthesized for each round. A central challenge is information synthesis: compressing salient facts and intermediate results; flagging uncertainty; and retiring stale information. Its effectiveness hinges on four meta-skills:

*verification* (detect and localize errors via self-judging and cross-candidate checks), *refinement* (use feedback/context to improve the artifact), *compression* (retain only past history via bounded summaries rather than replay), and *diversification* (exploratory variation to avoid consensus collapse).

**Learning to improve short-context iteration.** It is also of interest to teach the model a policy that effectively leverages this improvement operator. Standard RL training for reasoning models typically optimizes a single, long chain-of-thought conditioned on the prompt, with reward on the final answer (Shao et al., 2024; Guo et al., 2025). **PDR**, by contrast, comprises multiple short iterations that read a bounded summary, write a refinement, and re-synthesize a fresh summary. This creates a train-test mismatch in the information flow (short updates vs. one long trace). To make sure training is consistent with deployment, we optimize an objective that unrolls the operator itself during training: sample $M$ short drafts, distill them into a compact summary, and condition on the prompt plus that summary to produce a refined attempt. We use verifiable rewards to supervise the end-to-end computation. This objective narrows the train–test gap.

**Results.** On math tasks, iterative pipelines surpass single-pass baselines at matched sequential budgets; with shallow **PDR** delivering the largest gains (e.g., +11% on AIME 2024,

---

[1]Such LLMs fall within a traditional framework of *randomized space-bounded computation*. Computational complexity theory (Arora & Barak, 2007) shows it is capable of surprisingly powerful reasoning, such as determining connectivity of much larger graphs that cannot even fit the working memory; see Section D.

+9% on AIME 2025, +6% on LiveCodeBench, and +8% on GPQA). Making training consistent with inference, via an operator-consistent RL objective that optimizes the same read/write interface used at test time yields further improvements (e.g., $\sim 5\%$ on AIME 2024 and AIME 2025 when mixing standard and operator RL). These findings suggest that iteration with short contexts and compact summaries can substitute for long traces while holding latency fixed.

## 2. Background & Related Work

**Overview.** This section situates our work among several threads that seek to scale test-time reasoning: long-trace chains of thought and self-consistency; self-improvement and debate; structured search (trees/graphs of thoughts); multi-trace selection and aggregation; turning compute into supervision; memory/summarization; adaptive multi-turn RL; and learning-to-search/planning. We view these through a unified lens: *inference as a round-wise improvement operator under explicit budgets*; holding the per-call *sequential* budget $B_{seq}$ fixed while varying *total* compute $B_{total}$. We do not claim the micro-primitives themselves are new: parallel sampling, aggregation and selection (Wang et al., 2023; Fu et al., 2025; Zhao et al., 2025), sequential revision, critique-revise-verify and debate (Madaan et al., 2023; Gou et al., 2023; Shinn et al., 2023; Du et al., 2023b), structured search (Yao et al., 2023; Besta et al., 2024), and summarization/memory (Wu et al., 2025; Yang et al., 2024) are all well explored. Our contribution is to (i) formalize these pieces within a single round-wise operator (**SR**, **PDR**), (ii) analyze compute-normalized depth via shallow parallel rounds and distillation at matched $B_{seq}$ while varying $B_{total}$, and (iii) make training consistent with deployment via *operator-consistent RL*. The text below organizes prior work along these axes and clarifies similarities and differences.

**Test-time reasoning with long traces.** Eliciting step-by-step "chains of thought" improves accuracy on multi-step tasks (Wei et al., 2022). Recent "reasoning" models (e.g., OpenAI *o1*, Deepseek *r1*) explicitly trade more test-time thinking for better results, increasing tokens and latency (Jaech et al., 2024; Guo et al., 2025). Sampling multiple traces and aggregating answers (self-consistency) further boosts performance but scales cost with the number of samples (Wang et al., 2023). Our approach targets a complementary design point: keep each call short while letting evidence accumulate across rounds via a bounded, re-synthesized summary.

**Self-improvement.** A growing line of work lets models critique and refine their own outputs: *Self-Refine* alternates self-feedback and revision (Madaan et al., 2023); *Reflexion* maintains textual memory to guide subsequent attempts (Shinn et al., 2023); *CRITIC* verifies with tools and revises accordingly (Gou et al., 2023). Multi-agent debate improves factuality and reasoning via argumentation and adversarial checking (Irving et al., 2018; Du et al., 2023b). Our operator shares the self-improvement spirit but constrains per-call context by using a round-wise, re-synthesized compact state $C^{(r)}$ instead of replaying full histories. *Compute as Teacher* (CaT) synthesizes a single reference from parallel rollouts and optimizes the policy toward it, converting extra inference compute into reference-free supervision. Our focus differs: we use parallelism within **PDR** to explore and distill at inference time, and analyze compute explicitly under fixed $B_{seq}$ while varying $B_{total}$.

**Multi-trace selection and aggregation.** Confidence-aware test-time scaling generates multiple traces in parallel and filters or re-weights them with model-internal confidence, improving the accuracy–compute trade-off without extra training (Fu et al., 2025). Another line trains an *aggregator* to select or combine solutions using RL rather than majority vote or reward-model ranking (Zhao et al., 2025). In contrast, we cast inference itself as *round-wise operators* (**SR**, **PDR**), with aggregation as one of the meta-cognitive abilities necessary to improve the performance, and propose **PDR** RL to reduce the train-inference mismatch. Zheng et al. (2025) propose *Parallel-R1*, an RL framework that instills parallel thinking through a progressive curriculum: supervised fine-tuning on easier prompts to bootstrap the behavior, followed by RL on harder problems. On math benchmarks (MATH, AMC23, AIME), Parallel-R1 improves over a sequential-thinking RL baseline and shows that parallel thinking functions as a mid-training exploration scaffold before aiding verification in later stages.

**Structured search beyond single chains.** Prompting schemes structure exploration explicitly: *Tree of Thoughts* (ToT) expands and evaluates branches of reasoning (Yao et al., 2023); *Graph of Thoughts* generalizes to arbitrary thought graphs (Besta et al., 2024); *Least-to-Most* decomposes problems into subproblems solved in sequence (Zhou et al., 2022). These methods typically grow tokens with breadth/depth or rely on long contexts to carry intermediate state. In contrast, **PDR** concentrates exploration within a round, then distills to a bounded $C^{(r)}$, preventing unbounded context growth.

## 3. LLMs as Improvement Operators

### 3.1. Problem setting and notation

We consider tasks $x$ (e.g., a math problem) and aim to produce a high-quality final artifact $s_{final}$ (solution, proof, or program) under a given token budget. Let $\mathcal{M}_\theta$ denote a (frozen or trainable) LLM used as an *improvement operator*. Given a current artifact $s_t$ (single generation or set of generations) and a compact textual workspace $C_t$, the model

proposes a refinement:

$$s_{t+1} \leftarrow \mathcal{M}_\theta(x, s_t, C_t). \tag{1}$$

The workspace $C_t$ is a bounded summary ($|C_t| \leq \kappa$ tokens) meant to capture agreements, contradictions, intermediate results, and open subgoals.

**Read-write-compress cycle.** Each step (i) reads the current workspace $C_t$, (ii) writes a refined artifact $s_{t+1}$ via $\mathcal{M}_\theta$, and (iii) compresses back into a bounded workspace for the next step using a synthesis operator $\mathcal{D}$:

$$C_{t+1} \leftarrow \mathcal{D}(x, s_{t+1}), \qquad |C_{t+1}| \leq \kappa. \tag{2}$$

**Token budgets.** We evaluate every method under two budgets, $B_{\text{seq}}$, which represent the latency proxy (tokens along the accepted path) and $B_{\text{total}}$, which represents the compute/cost proxy (tokens from all model calls, including discarded branches). We report accuracy as a function of both axes and match baselines per axis (e.g., equal $B_{\text{seq}}$ for latency-controlled comparisons).

### 3.2. Operator Instantiations

We study two short-context iterative refinement pipelines.

#### 3.2.1. SEQUENTIAL REFINEMENT (**SR**; DEPTH OVER A SINGLE CANDIDATE).

We set $C_t \equiv \varnothing$ for all $t$ and iteratively improve a single artifact for $R$ rounds:

$$s_{t+1} \leftarrow \mathcal{M}_\theta(x, s_t, \varnothing), \quad t = 0, \ldots, R-1, \qquad s_{\text{final}} = s_R. \tag{3}$$

**SR with compact workspace.** In **SR**, no explicit workspace is provided. We also evaluate a variant that inserts an error analysis step between rounds: rather than directly refining the previous answer, the model first identifies and explains flaws in the current solution, then generates a revised solution. These notes act as a transient, local workspace at each round.

#### 3.2.2. PARALLEL-DISTILL-REFINE (**PDR**; ROUND-WISE WORKSPACE)

We do not maintain a persistent memory. Instead, for rounds $r = 1, \ldots, R$; we sample $M_r$ drafts (Parallel) conditioned on the current bounded summary, then re-synthesize (Distill) a fresh bounded summary for the next round:

$$\text{(Parallel)} \quad S^{(r)} = \left\{ s_i^{(r)} \leftarrow \mathcal{M}_\theta(x, C^{(r-1)}) \right\}_{i=1}^{M_r}, C^{(0)} = \varnothing, \tag{4}$$

$$\text{(Distill)} \quad C^{(r)} \leftarrow \mathcal{D}(x, S^{(r)}), \qquad |C^{(r)}| \leq \kappa. \tag{5}$$

We enforce single generation in last round $M_R = 1$; which is returned as $s_{\text{final}}$. The summary is round-wise and non-persistent: earlier text is not replayed, preventing growth in per-call context.

**Why a round-wise summary?** Replay of all prior attempts scales linearly with steps and reintroduces long-context failure modes. Re-synthesizing $C^{(r)}$ from the current drafts keeps the memory *bounded* ($|C^{(r)}| \leq \kappa$) and focuses each round on the most recent and informative evidence.

**Constructing the compact summary** $C^{(r)}$**.** We consider several practical instantiations of the distillation operator $\mathcal{D}$, all obeying $|C^{(r)}| \leq \kappa$:

- **Global summary:** Produce a single shared $C^{(r)}$ that captures agreements, contradictions, derived facts, unresolved subgoals, and next actions. This emphasizes verification and comparison while retiring stale or contradicted information.

- **Extractive top-$k$ evidence (shared):** Instead of free-form text, select the $k$ solutions from $S^{(r)}$ as the workspace itself, trading compression for higher fidelity to the best evidence.

- **Random-$k$ / bootstrapped workspaces:** For the next round, construct multiple small workspaces by randomly sampling $k$ solutions per generation. This injects diversity and mitigates premature consensus while keeping each workspace small.

**Budgets.** Tokens used for **Parallel**, **Distill**, and **Refine** contribute to $B_{\text{total}}$. The reported latency $B_{\text{seq}}$ only counts the tokens on the accepted generate→distill→refine path for the final output.

### 3.3. Operator-Consistent Training

The previous sections treat $\mathcal{M}_\theta$ as frozen and rely purely on prompting/orchestration. We now make sure training is consistent with deployment/inference by optimizing the model under the same short-context, iterative interface used at test time.

**Motivation.** Most RL for reasoning LLMs optimizes a single, long chain-of-thought trajectory. If inference instead runs multiple short passes with a compact workspace $C$, this creates a train-test mismatch. We remove this mismatch by mixing two training modes: (i) standard long-trace optimization, and (ii) *operator rollouts* that execute the generate→distill→refine interface under short contexts.

**Base Algorithm.** For the baseline RL, we use the CISPO objective from Minimax-M1 (Li et al., 2025). For a given prompt $x$, the generator $\pi(\cdot \mid \theta_{\text{old}})$ generates $G$ rollouts $\{o_{i=1}^G\}$ using the old policy $\theta_{\text{old}}$. Automated checkers like

sympy (Meurer et al., 2017) or math-verify[2] are used to assign scalar rewards $r_i$ ($\pm 1$) to each of the rollouts. CISPO combines the group-normalized advantage from GRPO (Shao et al., 2024) with REINFORCE (Williams, 1992) to achieve the following objective:

$$\mathcal{J}_{\text{CISPO}}(\theta) = \underset{x \sim \mathcal{D}, \{o_i\}_{i=1}^G \sim \pi(\cdot|x;\theta_{\text{old}})}{\mathbb{E}} \left[ \frac{1}{\sum_{i=1}^G |o_i|} \sum_{i=1}^G \sum_{t=1}^{|o_i|} \text{obj} \right] \tag{6}$$

$$\text{obj} = \text{sg}(r_{i,t}(\theta)) A_i \log(\pi(o_{i,t}|x, o_{i,<t}; \theta)) \tag{7}$$

where $A_i = \frac{r_i - \text{mean}(\{r\}_{j=1}^G)}{\text{std}(\{r\}_{j=1}^G)}$ is the advantage, sg is the stop-gradient operation, and $r_{i,t}(\theta)$ is computed using the asymmetric clipping from Yu et al. (2025) as follows:

$$r_{i,t} = \text{clip}\left( \frac{\pi(o_i \mid x, o_{i,<t}; \theta)}{\pi(o_i \mid x, o_{i,<t}; \theta_{\text{old}})}, 1 - \epsilon^-, 1 + \epsilon^+ \right) \tag{8}$$

where $\frac{\pi(o_i|x, o_{i,<t}; \theta)}{\pi(o_i|x, o_{i,<t}; \theta_{\text{old}})}$ is the importance-sampling (IS) weight. Additionally, we add an SFT loss (negative log-likelihood) similar to Seed et al. (2025) on rollouts which lead to positive rewards. The final training objective becomes:

$$\mathcal{J}(\theta) = \mathcal{J}_{\text{CISPO}}(\theta) + \alpha \cdot \mathcal{J}_{\text{SFT}}(\theta) \tag{9}$$

where $\alpha$ is set to a small value like $0.1$ in our experiments. The addition of this SFT loss boosts the utilization of positive rollouts and enforces better verification behavior in model training.

**Data mixture.** At each update, draw a mini-batch $\mathcal{B} = \{x_i\}_{i=1}^N$ and split it evenly into two sub-batches $\mathcal{B}_{\text{trace}}$ and $\mathcal{B}_{\text{op}}$ with $|\mathcal{B}_{\text{trace}}| = \lfloor N/2 \rfloor$ and $|\mathcal{B}_{\text{op}}| = \lceil N/2 \rceil$. We train on $\mathcal{B}_{\text{trace}}$ with a standard long-trace objective $\mathcal{J}_{\text{trace}}(\theta)$, and on $\mathcal{B}_{\text{op}}$ with *operator rollouts* under short context, yielding $\mathcal{J}_{\text{op}}(\theta)$. The per-step objective is the average of the two:

$$\mathcal{J}_{\text{train}}(\theta) = \tfrac{1}{2} \mathcal{J}_{\text{trace}}^{\mathcal{B}_{\text{trace}}}(\theta) + \tfrac{1}{2} \mathcal{J}_{\text{op}}^{\mathcal{B}_{\text{op}}}(\theta), \tag{10}$$

where $\mathcal{J}_{\text{trace}}^{\mathcal{B}_{\text{trace}}}$ and $\mathcal{J}_{\text{op}}^{\mathcal{B}_{\text{op}}}$ denote the losses computed on their respective sub-batches. Other ratios are possible; we use a 1:1 split in our experiments.

**Mode A: Standard long-trace optimization.** Given $x$, sample a single, long trajectory $s_{1:T} \sim \mathcal{M}_\theta(x)$ and optimize a conventional RL verifiable signal (e.g., a rule based verifiable reward for math problems). This preserves the model's ability to reason in extended traces when available.

**Mode B: Operator rollouts under short context.** We roll out the same interface used at test time but with one round for stability and cost.

*(i) Parallel-Distill-Refine (**PDR**; one-round rollout).*

1. Generate $M$ parallel generations (reasoning traces, solutions) conditioned on an empty summary:

$$S = \{ s_i \leftarrow \mathcal{M}_\theta(x, C^{(0)} = \varnothing) \}_{i=1}^M.$$

2. Distill to a bounded, round-wise summary $C$.

3. Refine a single candidate conditioned on $C$: $\tilde{s} \leftarrow \mathcal{M}_\theta(x, s_j, C)$.

**Why one round during training?** Rolling out a single **PDR** round (with $M$ early drafts, distillation to $C$, and a single refinement) captures the key interface while controlling $B_{\text{total}}$ and stabilizing RL. At inference we can run multiple rounds ($R > 1$) using the same operator.

Our datamix preserves competence on long traces while teaching the model to reason across short iterations. **PDR** is emulated by one-round of parallel→distill→refine rollout where the model observes $(x, C)$ and is optimized with a verifiable reward on the final solution trace.

## 4. Experiments

In this section, we compare the Sequential refinement (**SR**) and Parallel-Distill-Refine (**PDR**) operators against long chain-of-thought baselines under a budget-aware protocol. We measure accuracy with symbolic verifiers like sympy (Meurer et al., 2017) and math-verify[3]. Additionally, we report the results as functions of both the sequential budget $B_{\text{seq}}$ (latency proxy along the accepted path) and the total budget $B_{\text{total}}$ (all tokens across calls).

We try to answer the following four research questions through our experiments:

- **RQ1:** Can short-context iterations outperform long traces by comparing {**SR**, **PDR**} to long-trace CoT at matched $B_{\text{seq}}$ and $B_{\text{total}}$.

- **RQ2:** Figuring out the best distillation strategy for producing $C^{(r)}$ by comparing three $\mathcal{D}$ variants: global summary, extractive top-$k$, and random-$k$ bootstraps.

- **RQ3:** Identifying the effect of the verification ability of a given model on the final performance.

- **RQ4:** Whether operator-consistent training shifts the Pareto-Frontier. We compare a operator-consistent + standard RL with standard single-trace RL (Sec. 3.3).

---

[2]https://github.com/huggingface/Math-Verify

[3]https://github.com/huggingface/Math-Verify

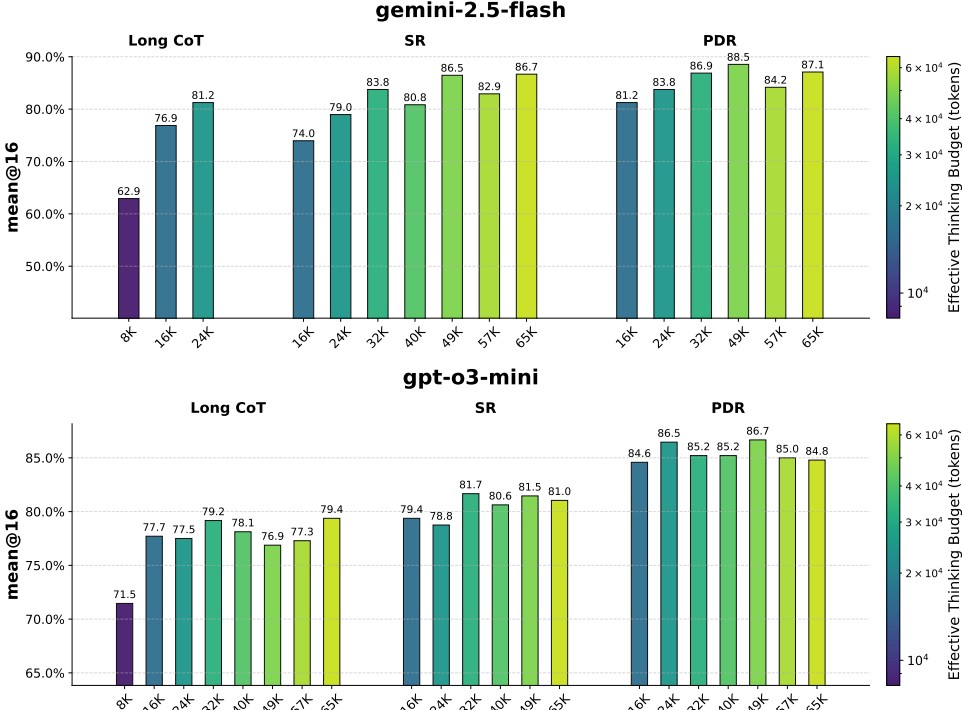

*Figure 2.* **AIME 2024: Iterative improvement beats single-pass long-CoT at matched sequential budgets.** The $x$-axis reports $B_{\text{seq}}$: the thinking tokens consumed along the accepted path of the iterative chain, plus any distilled summary that conditions the next step. Tokens spent on unused parallel proposals are excluded, so $B_{\text{seq}}$ serves as a latency proxy. At comparable $B_{\text{seq}}$, both **SR** and **PDR** outperform the single-pass long CoT baseline, with **PDR** yielding the largest gains by converting additional total compute (via parallelism) into accuracy without increasing per-call context.

### 4.1. Experiments to understand **SR** and **PDR**

**Setup.** We evaluate **SR** and **PDR** as inference-time operators on math problems. Given a prompt $x$, the model produces a thinking trace and a final solution. The thinking spans, delimited by `<think>...</think>` are stripped out and only the self-contained solutions are used to build the inputs for subsequent rounds. We evaluate on various reasoning benchmarks: AIME 2024 and AIME 2025 (AoPS, 2025), LiveCodeBench (2025 split) (Jain et al., 2024), and GPQA (Rein et al., 2024). We report the accuracy computed over 16 independent generations - mean@16.

**Models and inference budgets.** We evaluate `o3-mini` ("medium" reasoning effort) (OpenAI, 2025), `gemini-2.5-flash` (Comanici et al., 2025), and GPT-OSS series (`GPT-OSS 20B` and `GPT-OSS 120B`) (Agarwal et al., 2025). For all models except `o3-mini`, we vary the thinking budget from 8,192 to 24,576 tokens (its maximum), and reserve an additional 2,048 tokens for the final solution. Because `o3-mini` does not expose a separate thinking budget, we vary its maximum generation length from 10,240 to 26,624 tokens to match the same total allowance (assuming 8,192–24,576 thinking tokens plus 2,048 solution tokens). Both operators (**SR** and **PDR**)

are compared at matched per-call sequential budgets $B_{\text{seq}}$ (latency proxy) while allowing different total token budgets $B_{\text{total}}$ via parallelism. All runs use temperature $= 1.0$ and `top-p` $= 1.0$.

**RQ1: Do short-context iterations beat long traces at matched latency?**

**Sequential Refinement (SR).** For the **SR** operator, we run `o3-mini` and `gemini-2.5-flash` for thinking budgets $B \in \{8192, 16384, 24576\}$ and refinement rounds $r \in \{1, \dots, 6\}$. The prompt template is given in §B.1.

**SR with a local workspace.** We also evaluate a variant of **SR** that inserts a brief, local workspace between refinements: the model first performs *error analysis*: identifying and explaining flaws in the current solution, and then generates a revised solution conditioned on these notes. All other settings (prompts, budgets) match standard **SR** for a fair comparison. As shown in Table 3, this augmentation is effective for `o3-mini` but not for `gemini-2.5-flash`.

**Parallel-Distill-Refine (PDR).** We evaluate **PDR** under a fixed thinking budget $B$ using three settings. These settings differ by number of rounds, number of parallel generations in each round, and selecting the $k$ candidate solu-

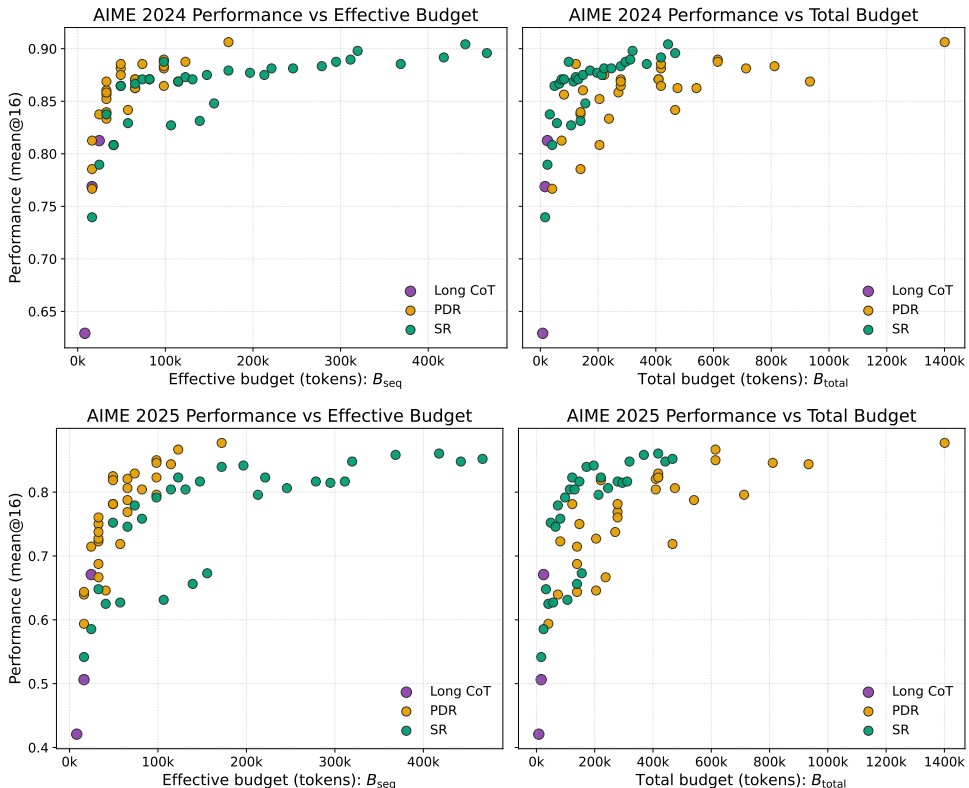

*Figure 3.* **Token Budgets comparison:** We plot all the different configurations for Long CoT, **SR** and **PDR** operators for both $B_{\text{seq}}$ and $B_{\text{total}}$ token budgets for `gemini-2.5-flash`. For $B_{\text{seq}}$, **PDR** forms the Pareto-frontier and gives consistent gains over Long CoT and **SR**. However, for $B_{\text{total}}$, **SR** forms the pareto-frontier because there are no parallel drafts involved so no generations are discarded.

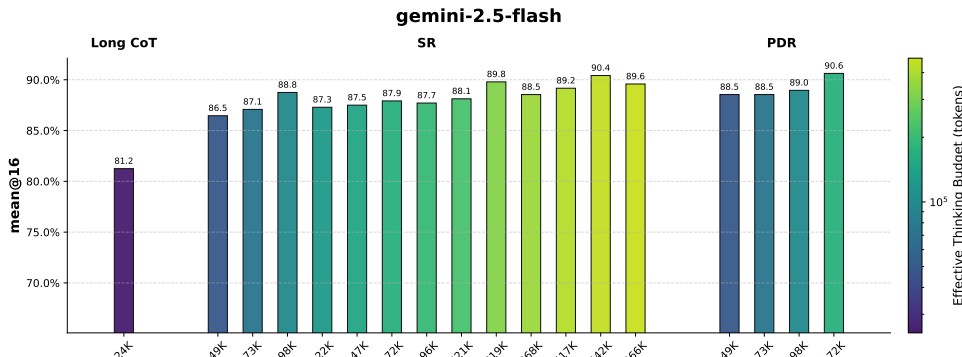

*Figure 4.* **AIME 2024: Long CoT, SR, and PDR at thinking budget of 24576.** The $x$-axis reports $B_{\text{seq}}$: the thinking tokens consumed along the accepted path of the iterative chain, plus any distilled summary that conditions the next step. Tokens spent on unused parallel proposals are excluded, so $B_{\text{seq}}$ serves as a latency proxy. At a token matched budget of $442k$ tokens, **SR** has a score of 90.4 but $B_{\text{seq}}$ of $442k$, whereas **PDR** has a score of 90.6 but $B_{\text{seq}}$ of $172k$ tokens.

tions to carry forward via textual workspace: $g = [8]$, $k=4$; $g = [16, 8]$, $k=4$; and $g = [16, 8, 4]$, $k=2$. Here $g = [g_1, \ldots, g_r]$ specifies the number of parallel generations in each round, and $k \leq \min_d g_r$ is the number of candidates forwarded to the next round. For distillation (i.e., selecting the $k$ candidates to carry forward), we compare: Random-$k$ (uniform sampling per instance); Top-$k$ (model-graded)

where the same base model assigns a quality score to each candidate and we keep the top $k$ per instance (we report both a shared rubric and a per-instance grading prompt); and global-summary that aggregates all candidates using a summarization prompt. Refinement, selection, and summarization prompts are detailed in §B.2.

**Token-matched baselines.** In Figure 4 we sweep the depth

of **SR** to find the sequential budget $B_{\text{seq}}$ at which it matches **PDR**. Holding the total token budget fixed at $B_{\text{total}} = 442\text{k}$ (for **SR**, $B_{\text{total}} = B_{\text{seq}}$ since there is no parallelism), **PDR** attains the target accuracy with $B_{\text{seq}} = 172\text{k}$. To reach the same accuracy, **SR** is run for 17 rounds, consuming $B_{\text{seq}} \approx 442\text{k}$. Thus, at equal $B_{\text{total}}$, **PDR** achieves the same accuracy with a $2.57\times$ smaller sequential budget by converting parallel compute into accuracy without lengthening per-call context.

**Results.** Figures 2 and 8 report accuracy on AIME 2024 and AIME 2025 under the same effective token budgets $B_{\text{seq}}$. We observe consistent gains when moving from long chain-of-thought to **SR**, which continue when moving from **SR** to PDR. For `o3-mini` at an effective budget of 49k tokens with a per-call thinking budget of 16k, accuracy improves from 76.9 (Long CoT) to 81.5 (**SR**) and 86.7 (**PDR**), an absolute improvement of $+9.8$ percentage points over Long CoT. `gemini-2.5-flash` shows smaller deltas from **SR** to **PDR** than `o3-mini`, suggesting stronger intrinsic self-verification in `gemini-2.5-flash`. AIME 2025 exhibits similar trends. Results with PDR on additional models and benchmarks are shown in Table 4.

**RQ2: Which distillation (i.e., summarization) strategy works best?**

Table 1 studies the distillation operator $\mathcal{D}$ in **PDR** under a (fixed number of rounds, number of generations in each round) setting $g = [16, 8, 4]$ with $k = 2$ candidates per round. Across datasets and base models, *per-sample top-k* and *global-summary* selection consistently outperform *shared top-k* and *random-k*, and the margin widens as the thinking budget $B$ increases. The main exception is AIME 2025 with `o3-mini`, where global summary outperforms the alternatives. We hypothesize that `o3-mini`'s summarization is particularly effective at capturing cues from both correct and incorrect drafts, and these cues, when distilled, lead to stronger subsequent refinements.

**RQ3: How does the verification abilities effect the inference time performance ?**

**Oracle PDR analysis.** To understand the role of model verification within **PDR**, we intervene on the set of candidates admitted to the summary at each round. We use a three-round **PDR** with number of generations in each round as $[16, 8, 4]$ and top-$k$ selection ($k = 2$), and compare: (i) **Random-$k$:** choose $k$ candidates uniformly at random from the previous depth; (ii) **Oracle (Correct):** admit only correct candidates to the compact summary when available; (iii) **Oracle (Incorrect):** admit only incorrect candidates.

From Figures 6 and 7, we observe that injecting incorrect candidates (Oracle (Incorrect)) causes large drops for all models. The degradation is substantially larger for `o3-mini` than for `gemini-2.5-flash`, suggesting

stronger self-verification and recovery in the latter. The same trend holds across AIME 2024 and AIME 2025.

We further provide a detailed analysis on the mechanics of **PDR** and self-verification requirements to improve downstream performance in Section F.

## 4.2. Operator-consistent RL Training

**RQ4: Does operator-consistent training move the Pareto Frontier?**

Building on the above, we present an operator consistent RL training strategy. This also addresses a train-test gap where models are not explicitly trained to perform **PDR**.

**Training setup.** We train an 8B dense model similar to Llama-3 style architecture (Dubey et al., 2024). For warm-start supervised fine-tuning (SFT), we use GPT-OSS 120B (Agarwal et al., 2025) to generate synthetic traces for math and code prompts sampled from Polaris-53K (An et al., 2025) and DeepCoder (Luo et al., 2025), respectively. We run SFT for 8B tokens ($\sim$4 epochs). For RL training, we use the Polaris-53K dataset. Both SFT and RL datasets are decontaminated against AIME 2024/2025 (AoPS, 2025) and MATH-500 (Hendrycks et al., 2021). Further details and hyper-parameters are detailed in Section C.

**Baseline RL** As described in Section 3.3, we use the CISPO objective for RL post-training (Li et al., 2025). We set $\epsilon^- = 0.0$ and $\epsilon^+ = 5.0$, and remove "zero-variance" prompts from a given batch (Seed et al., 2025). We use forced interruptions (Hong et al., 2025; Yang et al., 2025) to control generation length from exploding after a thinking budget of $16,384$ tokens. We additionally keep a buffer of $2048$ tokens for the final solution, thus keeping a maximum generation length of $18432$. $32$ generations are sampled per prompt with a batch size of $32$, resulting in a global batch size of $1024$ generations per gradient step. Following (Liu et al., 2025b), we use a mini-batch size of $256$ and perform $4$ gradient updates per rollout step.

**Operator-consistent RL with PDR.** For training, we use the **PDR** operator with configuration (4 parallel generations, 1 round) $g = [4]; k = 2$, and use the training objective described in Equation (10) and make two changes to the baseline RL method above: (i) increasing the input prompt length from 2048 tokens to 10240 to allow for the compact workspace to be a part of the input, and (ii) mixing the standard RL and operator RL batches in the dataloader, keeping all other design choices the same. This setup allows to scale inference compute within the RL training.

**Results.** Table 2 summarizes the main results. The resulting model from each RL objective is evaluated for Long CoT generation and **PDR**. **PDR** RL improves over the baseline by $+3.34$ points on AIME 2024 and $+1.67$ points on

*Table 1.* **Effect of distillation operator** $\mathcal{D}$**:** We compare the effect on final performance by changing the distillation operator $\mathcal{D}$. Each table column reports accuracies on AIME 2024 / AIME 2025. We compare four choices: (i) *Global summary*: aggregate all candidates and synthesizes a single compact summary; (ii) *Per-sample top-k*: each downstream branch selects its own top-$k$ candidates as the summary; (iii) *Shared top-k*: a single set of top-$k$ candidates is shared as the summary across generations for next round; (iv) *Random-k*: each generation for next round receives $k$ candidates sampled uniformly at random for the summary. Overall, global summary and per-sample top-$k$ tend to perform best, with gains more pronounced at higher thinking budgets. For o3-mini on AIME 2025, global summary yields the largest improvement, suggesting strong summarization ability in o3-mini. We use $k = 2$ for these experiments.

| Budget | gemini-2.5-flash | | | | gpt-o3-mini | | | |
|---|---|---|---|---|---|---|---|---|
| | Global | PS top-$k$ | Shared top-$k$ | Random-$k$ | Global | PS top-$k$ | Shared top-$k$ | Random-$k$ |
| 8192 | 83.13 / 66.88 | 83.75 / 71.88 | 84.17 / 70.21 | 83.33 / 66.67 | 86.04 / 82.92 | 84.79 / 76.04 | 85.00 / 76.67 | 82.50 / 71.25 |
| 16384 | 86.46 / 84.38 | 86.88 / 83.96 | 86.46 / 83.75 | 86.25 / 80.63 | 86.46 / 84.79 | 85.42 / 74.58 | 85.83 / 76.88 | 83.13 / 71.88 |
| 24576 | 88.75 / 87.71 | 90.63 / 85.00 | 87.71 / 85.42 | 88.13 / 79.58 | 85.42 / 83.54 | 85.21 / 77.92 | 85.00 / 75.42 | 82.29 / 72.08 |

*Table 2.* **Operator RL results:** Comparison of RL training objectives on AIME 2024/2025 at matched sequential budget $B_{\text{seq}} = 65{,}536$ tokens using a dense 8B model. Mixing standard RL with operator-consistent RL (Op-RL) yields consistent gains for iterative inference operators such as `PDR` while preserving performance on the Long CoT baseline. Op-RL can also be applied as a continual RL to the existing baseline RL checkpoint.

| Model | AIME 2024 | | AIME 2025 | |
|---|---|---|---|---|
| | Long CoT | `PDR` | Long CoT | `PDR` |
| 8B SFT Policy | 47.50 | 62.92 | 35.00 | 47.50 |
| 8B Baseline RL | 67.50 | 75.83 | 59.58 | 65.83 |
| 8B `PDR` RL | 69.58 | 79.17 | 57.50 | 67.50 |
| 8B Continual `PDR` RL | 70.00 | 80.83 | 61.25 | 70.42 |

AIME 2025. With continual updates starting from a baseline RL checkpoint, additional `PDR` RL yields larger gains of +5.00 and +4.59 percentage points on AIME 2024 and AIME 2025, respectively. Additionally, we also observe marginal gains on Long CoT generations with `PDR` RL training. These results indicate that training with operator-consistent RL objectives reduces the mismatch between training and deployment, converting extra compute into accuracy without increasing the per-call sequential budget. We also do a profiling analysis in Section G to show that Long CoT and PDR have similar latencies at matched sequential budgets $B_{seq}$.

## 5. Conclusion

In this paper, we initiate the exploration of a broader design space around "long CoT." We study two operators in this design space, `SR` and `PDR` which give better accuracy compared to standard long CoT, while offering the benefit of smaller context size. Empirically, compact-memory iteration outperforms long-trace baselines at matched $B_{\text{seq}}$. `PDR` yields the largest gains (e.g., +11% on AIME 2024 and +9% on AIME 2025), showing that evidence accumulation via bounded summaries can substitute for long reasoning traces while holding latency fixed. Beyond inference orchestration, making sure that training is consistent with inference using an *operator-consistent RL* objective further improves performance (e.g., ∼ 5% on AIME 2024 and AIME 2025), suggesting that models can learn the meta-skills that make iteration effective. Iterative reasoning improves when diversity, verification, and refinement become reliably good; by

measuring and training these micro-skills directly, we can accelerate the gains of improvement operators under fixed latency budgets.

Promising future directions include learning to improve the synthesis operator $\mathcal{D}$ (trainable summaries), adaptive round and fan-out schedules conditioned on uncertainty (adaptive $top - k$), budget-aware controllers for allocating test-time compute, and tighter integration with verifiers and tool use. We also see value in scaling studies and cross-domain evaluations (reasoning, coding, and planning) to map when short-context iteration most benefits accuracy and latency.

## Impact Statement

This paper presents PDR, an improvement operator for LLMs for both inference and training using Reinforcement Learning. If widely adopted, these methods could make high-accuracy reasoning more accessible by reducing effective latency and avoiding long-context failure modes, benefiting downstream applications like coding agents and interactive use cases.

Deploying these methods in high-stakes environments and out of distribution data can result in some reinforced bias on wrong model outputs as discussed in Section 4.1.

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

# A. Extended Related Work

**Multi-agent debate and compressed debate.** Debate-style methods have multiple LLM "agents" propose answers and iteratively read and critique one another, often improving robustness (Du et al., 2023a). Our *Parallel-Distill-Refine (**PDR**)* can be seen as a *compressed* debate: treat a round's diverse drafts as agent outputs, but instead of replaying full transcripts, distill them into a bounded $C^{(r)}$ that conditions the next round. This preserves cross-agent scrutiny while controlling per-call context and both $B_{seq}$ and $B_{total}$.

**Compact summaries vs. persistent memory.** Agent systems often add external memory or retrieval to persist context across sessions/tasks; thought buffers and memory-augmented agents exemplify this direction (e.g., Buffer-of-Thoughts) (Yang et al., 2024). We instead use *non-persistent*, round-wise summaries. This choice keeps prompts short and reduces long-context failure modes (Liu et al., 2023). Concurrently, Wu et al. (2025) introduce ReSum, a web-agent paradigm that periodically summarizes growing interaction histories into compact states and pairs this with ReSum-GRPO so agents learn summary-conditioned reasoning. Our work differs in focusing on multiple inference-time operators instead of summary (**SR**, **PDR**), alongside operator-consistent RL aligned to the round-wise interface. Didolkar et al. (2025) introduce *Metacognitive Reuse*, extracting recurring reasoning into concise, named behaviors that reduce token usage and can be distilled via SFT. This is complementary to our within-instance **PDR**: rather than compressing cross-instance procedures, we parallelize and distill drafts per instance.

**Adaptive multi-turn training.** Unary-feedback multi-turn RL ("try again") trains models to revise across rounds improving multi-turn accuracy with minimal supervision (Liu et al., 2025a). *Exploratory Iteration* (ExIt) leverages the recurrent structure of self-improvement by performing multi-step refinement at test time while training emphasizes the most informative single-step iterations (Jiang et al., 2025). **SR** is similar to "try again" iterative interface.

**RL with on-policy tree search.** Hou et al. (2025) introduce *TreeRL*, which integrates on-policy tree search into RL for LLM reasoning, improving exploration and providing dense, process-level rewards (Xie et al., 2024) compared to independent chain sampling with outcome-only supervision. TreeRL thus exemplifies a *training-time, search-augmented* improvement operator. In contrast, our operator-consistent RL trains in the same iterative interface used at inference: we unroll a single round (akin to a shallow tree expansion) but optimize with outcome supervision without deep tree search. A promising direction is to incorporate process-level rewards into this operator-consistent setting and study their impact on the test-time performance of round-wise operators (**SR**, **PDR**).

**Budget-aware evaluation and test-time compute.** Recent work argues for comparing methods at matched compute budgets and reporting token usage, and not just accuracy (Wang et al., 2024). Our protocol reports sequential budget $B_{seq}$ (latency proxy along the accepted path) and total budget $B_{total}$ (all tokens, including discarded branches), enabling apples-to-apples comparisons among single-pass, long-trace, sampling-heavy, and iterative pipelines.

**MCTS, Learning to search and amortizing search.** AlphaGo, AlphaZero, and MuZero couple a learned policy/value with an expensive test-time search (e.g., MCTS) that serves as an improvement operator; the outputs of search are then distilled back into the network, amortizing future search cost (Silver et al., 2016; 2017; Schrittwieser et al., 2020). *Expert Iteration* formalizes this loop as policy improvement via planning followed by supervised or RL updates toward the planner's targets (Anthony et al., 2017). Earlier "learning to search" work in structured prediction similarly alternates local rollouts with policy updates (e.g., SEARN, DAgger) (Daumé III & Langford, 2006; Ross et al., 2011). Our setting is analogous in spirit but distinct in mechanics: we operate in the *textual reasoning* regime with round-wise operators (**SR**, **PDR**) that keep the per-call sequential budget $B_{seq}$ small, optionally raising total compute $B_{total}$ via parallel drafts. Operator-consistent RL then amortizes this improvement procedure into the model weights.

**Global workspace and modular coordination.** Our compact, round-wise summary $C^{(r)}$ is conceptually related to the *shared global workspace* proposed by Goyal et al. (2022), which enables coordination among neural modules through a small communication bottleneck (inspired by Global Workspace Theory (Baars, 2005; Shanahan, 2006)). In contrast, our workspace is textual, re-synthesized at each round rather than persisted, and used as an inference-time operator. Thus, we borrow the coordination intuition while avoiding long-context replay and architectural changes.

# B. Prompts

## B.1. Sequential Refinement

**Refinement Prompt**

```
Solve the following math problem efficiently and clearly. Please reason step
    by step, and put your final answer within $\\boxed{answer}$.

Where [answer] is just the final number or expression that solves the problem
    .

Problem: {{ problem }}

Here is an example candidate response wrapped in angle brackets:

<model_response>
# Solution response from previous turn
</model_response>

Treat the response as unverified; and come up with a better answer without
    starting from scratch.
```

## B.2. Parallel-Distill-Refine

**Refinement Prompt (Non-summary)**

```
Solve the following math problem efficiently and clearly. Please reason step
    by step, and put your final answer within $\\boxed{answer}$.

Where [answer] is just the final number or expression that solves the problem
    .

Problem: {{ problem }}

Here are some candidate responses, each wrapped in angle brackets:

<model_response_1>
# Solution response from previous round
</model_response_1>

...

<model_response_k>
# Solution response from previous round
</model_response_k>

Treat these responses as unverified; and use these responses to come up with
    a better answer without starting from scratch.
```

**Refinement Prompt (Summary)**

```
Solve the following math problem efficiently and clearly. Please reason step
    by step, and put your final answer within $\\boxed{answer}$.

Where [answer] is just the final number or expression that solves the problem
    .

Problem: {{ problem }}

Here is a summary of the reasoning traces by a few other solvers:

<summary>
# Summary of all solutions from the previous iteration
</summary>

Treat the summary as unverified; and use the summary as context to come up
    with an answer.
```

## C. Training Setup

We run a small SFT on the pre-trained 8B base model using a batch size of 2M tokens, max sequence length of 32768, and a learning rate of $2 \times 10^{-5}$ using the AdamW optimizer (Loshchilov & Hutter, 2017) on 32 H100 GPU nodes for approximately 4 epochs and 8B tokens in total. For RL, we use a constant learning rate of $5 \times 10^{-7}$, AdamW optimizer (Loshchilov & Hutter, 2017) with $\epsilon = 10^{-15}$, weight decay of 0.01, and a linear warmup of 100 steps. We use 80 Nvidia H200 GPUs for the baseline RL run with a 64/16 generators/trainers split and 288 H200 GPUs for **PDR** and continual **PDR** RL with a 256/16 generators/trainers split to parallelize inference during rollout generation. We run all RL training for 800 steps. All evaluations are done with a temperature and `top-p` value of 1.0.

## D. Complexity of Space-bounded computation

Our work focused on language models that emit reasoning traces that are longer than the context size. We sketch similarity to the setting *space-bounded computation* which is formally studied in computational complexity theory. Since LLMs have probabilistic output (unless if temperature is set to 0) the fixed-context LLM considered in the paper is most similar to randomized space-bounded machine.

The most interesting result about randomized space-bounded machines is that if the input contains a graph of $N$ vertices, then the randomized machine can determine connectivity of the $N$-vertex graph even though it only has $\mathcal{O}(\log N)$ space.

Furthermore, imagine that the graph of size $N$ is a knowledge-graph whose local structure is known to the space-bounded machine. Specifically, given vertex indices $i, j$ the space-bounded machine is able to determine whether edge $\{i, j\}$ exists in the graph. Then the machine does not need access to the full graph tape at all! It can do a random walk through the graph "in its mind" to determine connectivity. This is the closest setting to ours, whereby seemingly complex reasoning-connectivity of an $N$-node graph can be carried out in less than $\mathcal{O}(N)$ space.

## E. Additional Results

### E.1. Oracle

Similar to Figure 6, we observe that having incorrect solutions in the context workspace can heavily degrade performance, and this effect is more noticeable for `o3-mini`.

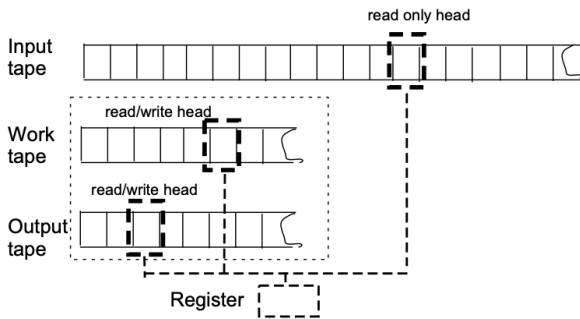

*Figure 5.* Space-bounded Turing Machine [Figure from *Computational Complexity* by Arora and Barak, 2007]. The input has size $N$ and the machine has read-only capability for the input. A special "tape head" can be moved over the input to read bits from it. The amount of working memory (read/write/erase) for actual computation has size $S(N)$ where $S(N) \geq \log N$.

*Table 3.* **SR operator variants**: Instead of just asking the model to refine the solution, we also ask the model to find and analyze errors in the solution followed by the correct solution. Error analysis followed by solution generation leads to better performance for `o3-mini` but not for `gemini-2.5-flash`.

| Model | Benchmark | Thinking Budget | **SR** | **SR**-*Error* |
|---|---|---:|---|---|
| gemini-2.5-flash | AIME 2024 | 24576 | 88.75 | 87.71 |
| gemini-2.5-flash | AIME 2025 | 24576 | 78.75 | 79.17 |
| gpt-o3-mini | AIME 2024 | 24576 | 80.83 | 82.08 |
| gpt-o3-mini | AIME 2025 | 24576 | 73.13 | 77.92 |

*Table 4.* **Long CoT vs PDR on all benchmarks**: We report the performance of `gemini-2.5-flash`, `o3-mini`, `GPT-OSS 20B`, and `GPT-OSS 120B` on reasoning benchmarks at matched sequential budgets $B_{seq}$ using both Long CoT and the PDR operator. For LiveCodeBench, we report results on the 2025 split, and use the diamond split for GPQA. Across all benchmarks, PDR consistently improves over the baseline, suggesting that its benefits extend to all reasoning domains.

| Model | AIME 2024 | | AIME 2025 | | LiveCodeBench | | GPQA (diamond) | |
|---|---|---|---|---|---|---|---|---|
| | Long CoT | **PDR** | Long CoT | **PDR** | Long CoT | **PDR** | Long CoT | **PDR** |
| `gemini-2.5-flash` | 81.25 | 88.54 | 67.08 | 82.50 | 55.96 | 61.25 | 75.98 | 82.32 |
| `gpt-o3-mini` | 79.17 | 86.67 | 73.54 | 82.92 | 48.90 | 58.61 | 73.30 | 76.26 |
| `GPT-OSS 20B` | 91.67 | 94.17 | 90.63 | 93.96 | 55.62 | 63.99 | 61.55 | 68.50 |
| `GPT-OSS 120B` | 95.00 | 97.08 | 92.50 | 95.21 | 63.12 | 69.71 | 69.92 | 72.29 |

*Figure 6.* **AIME 2024: Anchoring bias due to +ve and −ve examples:** With **PDR** we compare three selection policies for the summary: Random-$k$, Oracle-Incorrect (all $k$ candidates are incorrect), and Oracle-Correct (all $k$ candidates are correct), evaluated on both `gemini-2.5-flash` and `o3-mini`. Across all thinking budgets, admitting only incorrect candidates into the summary yields a pronounced drop in accuracy, whereas admitting only correct candidates improves over the Random-$k$ baseline. The degradation under Oracle-Incorrect is larger for `o3-mini` than for `gemini-2.5-flash`, indicating weaker self-verification in `o3-mini`.

### E.2. **SR** and **PDR** operators

AIME 2025 results using the two iterative improvement operators **SR** and **PDR** are presented in Figure 8. For sequential token budget of $49k$, the performance on `o3-mini` improves from 73.5 for Long CoT to 77.1 using **SR** operator and further to 82.9 using the **PDR** operator.

Additionally, we show two **SR** variant results in Table 3, where error analysis followed by solution generation leads to improvements on `o3-mini` without any affect on the sequential token budget $B_{\text{seq}}$.

## F. Mechanics of Improvement operator: Source of accuracy gain

Under the default **PDR** setting, `gemini-2.5-flash` misses 4 AIME 2024 questions. We probe how additional parallel compute affects the performance on these four AIME questions. We compare a 4-round schedule (less compute) to a 5-round, wider schedule (more compute). Accuracy (fraction correct over 16 seeds) changes as follows: Q1: $0.4375 \rightarrow 0.625$ (gain), Q2: $0.0625 \rightarrow 0$ (drop), Q3: $0.1875 \rightarrow 0.1875$ (no change), Q4: $0 \rightarrow 0$ (no change). In the high-compute setting (first round width $M_1{=}32$), number of correct drafts among 32 is: Q1: 3/32, Q2: 0/32, Q3: 3/32, Q4: 0/32. This breakdown clarifies how **PDR** can (or cannot) improve with additional rounds:

1. When a round-1 draft is correct, **PDR** improves if two things happen: (i) the correct evidence is carried into the summary $C^{(1)}$ (i.e., high recall in the distillation operator $\mathcal{D}$); and (ii) the refine step uses that evidence to update the answer. If $\mathcal{D}$ drops or down-weights the signal amid conflicting drafts, later rounds cannot exploit it. The Q1 gain from 4→5 rounds

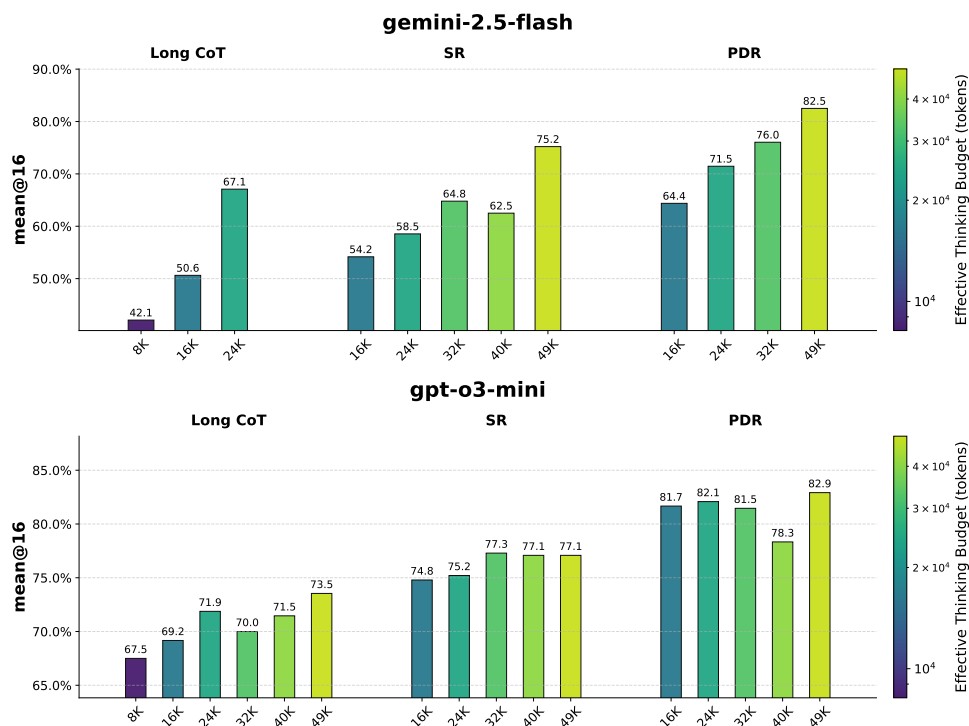

*Figure 7.* **AIME 2025: Anchoring bias due to $+$ve and $-$ve examples:** With `PDR` we compare three selection policies for the summary: Random-$k$, Oracle-Incorrect (all $k$ candidates are incorrect), and Oracle-Correct (all $k$ candidates are correct), evaluated on both `gemini-2.5-flash` and `o3-mini`. Across all thinking budgets, admitting only incorrect candidates into the summary yields a pronounced drop in accuracy, whereas admitting only correct candidates improves over the Random-$k$ baseline. The degradation under Oracle-Incorrect is markedly larger for `o3-mini` than for `gemini-2.5-flash`, indicating weaker self-verification in `o3-mini`.

*Figure 8.* **AIME 2025: Iterative improvement beats single-pass long-CoT at matched sequential budgets.** The $x$-axis reports $B_{\text{seq}}$: the thinking tokens consumed along the accepted path of the iterative chain, plus any distilled summary that conditions the next step. Tokens spent on unused parallel proposals are excluded, so $B_{\text{seq}}$ serves as a latency proxy. At comparable $B_{\text{seq}}$, both `SR` and `PDR` outperform the single-pass long CoT baseline, with `PDR` yielding the largest gains by converting additional total compute (via parallelism) into accuracy without increasing per-call context.

suggests both steps succeeded; the flat Q3 curve despite $3/32$ correct drafts points to a verification/refinement gap.

2. **Verification among many distractors (Q1/Q3).** Even when correct drafts are present, round-1 mixes a small number of

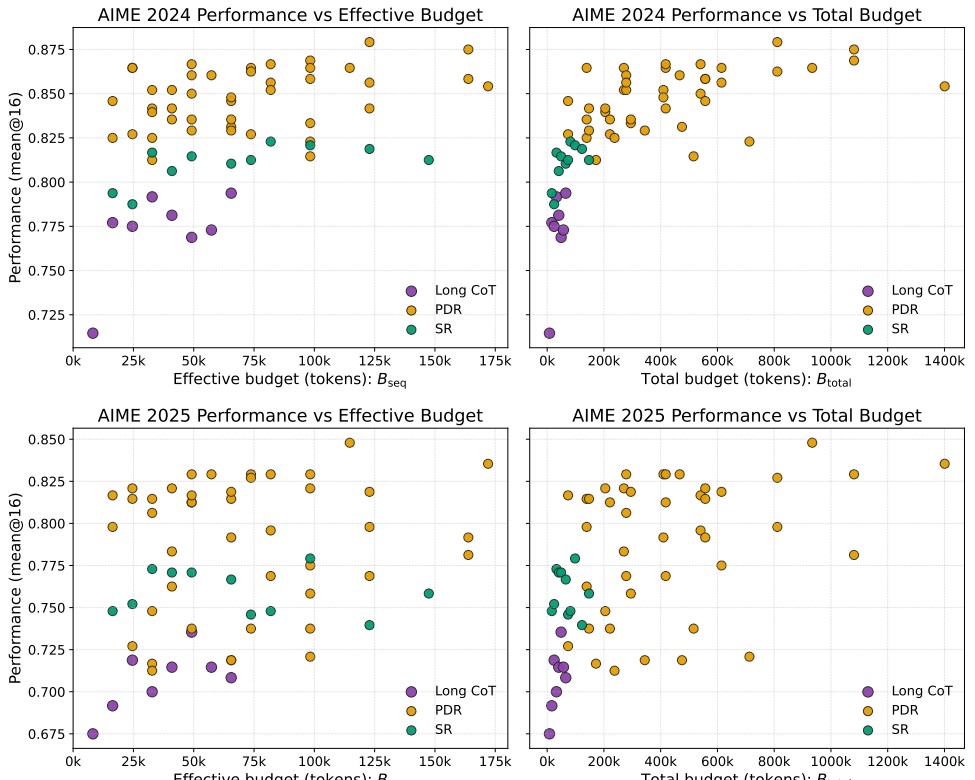

*Figure 9.* **Token Budgets comparison:** We plot all the different configurations for Long CoT, **SR** and **PDR** operators for both $B_{seq}$ and $B_{total}$ token budgets for `o3-mini`. For both $B_{seq}$ and $B_{total}$, **PDR** forms the pareto-frontier and gives consistent gains over Long CoT and **SR**.

correct drafts with many incorrect ones (here, 3 vs. 29). The summary must surface signals that distinguish correctness. This is where **PDR**'s distill step should act as a verifier-aware aggregator.

3. **Recovery when no draft is correct (Q2/Q4).** If round 1 has $0/32$ correct, the summary still needs to extract useful structure (partial progress, contradictions, eliminated avenues) that increases the chance of success in round 2+. The refine step should then expand diversity informed by these cues. The mild regression on Q2 with more compute is consistent with summary drift: the distillation over-weights a wrong pattern, and subsequent rounds reinforce it. The no-change on Q4 may suggest either a capability ceiling or the case that even more generations are required in round 1.

Overall, the analysis here points to some gaps in the core skills required to carry out **PDR** - verification, refinement, and diversity. Improving the model along each of these skills will not only improve **PDR** but also any situation where a multiplicative combination of these skills is required.

## G. Latency profiling on the 8B model

We do a simple inference profiling test using an 8B model on 1 H100 GPU for Long CoT and PDR. We use an input prompt length of 8192 tokens, batch size of 1 for Long CoT. We use an output generation length ranging from 1k-73k tokens and report the latencies in Table 6. For PDR, we use batched computation and set batch size of 8 for generation, and output generation length of 8k tokens. We do 5 measurements for each setting, and report the average latencies. As we can see in Table 6, a batched computation of 8 parallel branches (last row) takes 113.8501 - 101.3881 = 12.462 seconds more (for generation budget of 8192 tokens), but it's not significant given it's using 8x more FLOPs and generating 8x more total tokens. As we can observe in Table 7, for PDR, the first LLM call takes 113.8501 seconds and the second call takes 101.3881 seconds (because batch size=1 during refinement in PDR), so a total of 215.2382 seconds. PDR (row 3) is processing more total tokens compared to Long CoT (row 1), but we are able to leverage GPU parallelization to keep similar latencies and much higher accuracy for PDR compared to Long CoT for a fixed $B_{seq}$. If we consider $B_{total}$, PDR (row 3) latency is

*Table 5.* **PDR hard-case analysis for `gemini-2.5-flash`.** We compare a 4-round schedule (*less compute*: 16 generations in round 1) to a 5-round, wider schedule (*more compute*: 32 generations in round 1). Accuracies are fraction correct over 16 seeds. "Round-1 hits" counts how many of the 32 first-round drafts are already correct ((*more compute setting*)

| | Round-1 hits | Accuracy (over 16 seeds) | | $\Delta$ | Interpretation |
|---|---|---|---|---|---|
| Question | (correct/32) | Less compute | More compute | (More − Less) | |
| Q1 | 3/32 | 7/16 (0.4375) | 10/16 (0.6250) | +3/16 (+0.1875) | Gain |
| Q2 | 0/32 | 1/16 (0.0625) | 0/16 (0.0000) | −1/16 (−0.0625) | Drop |
| Q3 | 3/32 | 3/16 (0.1875) | 3/16 (0.1875) | 0 | Flat |
| Q4 | 0/32 | 0/16 (0.0000) | 0/16 (0.0000) | 0 | Flat |

*Table 6.* **Latency Profiling**: Comparison of latency on an 8B model while varying generation budgets and batch sizes. TTFT represents the time to first token (prefill time).

| Generation Length | Batch Size | TTFT (seconds) | Total Time (seconds) |
|---|---|---|---|
| 1024 | 1 | 0.1637 | 12.6808 |
| 2048 | 1 | 0.1640 | 25.2173 |
| 8192 | 1 | 0.1647 | 101.3881 |
| 16384 | 1 | 0.1714 | 202.7761 |
| 32768 | 1 | 0.1795 | 412.5801 |
| 65536 | 1 | 0.1976 | 830.8500 |
| 73728 | 1 | 0.2012 | 938.6954 |
| 8192 | 8 | 1.3354 | 113.8501 |

much lower and we still get better performance in PDR because the Long CoT (row 2) performance stagnates after a certain generation budget. There's a tradeoff between compute spent (FLOPs) and accuracy, and PDR is able to extract out more performance by spending a lot more compute. The affect on latency is minor because of GPU parallelization.

Additionally, this latency effect can be reduced further by deploying multiple model copies and reducing the batch size, which will decrease the minor latency gap between Long CoT and PDR while retaining the high performance of PDR compared to Long CoT.

*Table 7.* **Latency Profiling**: Long CoT vs PDR comparison.

| Configuration | Generation Budget | LLM Calls | $B_{seq}$ | $B_{total}$ | Latency (s) |
|---|---|---|---|---|---|
| Long CoT | 16384 | 1 | 16384 | 16384 | 202.7761 |
| Long CoT | 73728 | 1 | 73728 | 73728 | 938.6954 |
| PDR ($w = 8, k = [4]$) | 8192 | 2 | 16384 | 73728 | 215.2382 |

