# OpenReview forum: "Rethinking Thinking Tokens: LLMs as Improvement Operators"
_ICML.cc/2026/Conference — ICML 2026 regular_

### Official Review · Reviewer_pHZ1 · 2026-03-04

**Soundness:** 2
**Presentation:** 3
**Significance:** 2
**Originality:** 3
**Overall Recommendation:** 3
**Confidence:** 3

**Summary:**

The paper studies how to improve large language model reasoning at inference time by treating the model as an improvement operator over its own intermediate thoughts. Instead of generating a single long chain-of-thought, the model iteratively refines candidate solutions while keeping the context length bounded. The authors propose two main inference strategies: Sequential Refinement (SR) and Parallel-Distill-Refine (PDR). In Sequential Refinement, the model repeatedly updates a single draft solution across several short rounds, gradually improving it. In Parallel-Distill-Refine, the model first generates multiple diverse drafts in parallel, then distills useful information from these drafts into a compact textual workspace, and finally refines the solution conditioned on this workspace, repeating the process over multiple iterations. Experiments on reasoning benchmarks (e.g., AIME math tasks) show that these iterative strategies can outperform traditional long chain-of-thought reasoning while using a comparable token budget.

**Compliance With Llm Reviewing Policy:**

Affirmed.

**Final Justification:**

Overall, I find the paper interesting, but it still has notable limitations. Therefore, I maintain my decision.

**Key Questions For Authors:**

1. Have you tested SR or PDR on other reasoning domains?

2. The improvement operator is not monotonic, and a poor workspace can degrade the solution.Can this degradation probability/impact be bounded?

**Limitations:**

yes

**Strengths And Weaknesses:**

* Soundness:

The idea of moving beyond a single long chain-of-thought to an iterative refinement process is interesting, and the experiments show improved performance under the same token budget.

However, the empirical evaluation is limited, focusing mainly on AIME-style math problems, which makes it unclear how well the approach generalizes to other reasoning tasks. In addition, the improvement operator is not monotonic: the oracle (incorrect) experiment suggests that a poor workspace can degrade the answer. This may cause SR to amplify errors across iterations, and mitigating this issue in PDR may require additional filtering or judging steps, increasing computational cost.

* Presentation:

The paper is generally clear and well structured, and the description of SR and PDR is easy to follow. However, the connection drawn to randomized space-bounded computation is not very convincing and does not appear essential to the main contribution.

* Significance:
The paper addresses an important problem. However, the limited evaluation makes it difficult to assess the broader impact of the approach.

* Originality:

The work presents a novel perspective of viewing LLM reasoning as an iterative improvement process rather than a single reasoning trajectory. While the components build on existing ideas (e.g., CoT and self-refinement), the framework and the PDR strategy provide an interesting combination that could inspire future work on test-time scaling.

---

> ### Author Rebuttal · Authors · 2026-03-31
>
> We thank the reviewer for their comments and address the weaknesses/questions along with additional experiments below:
>
> > However, the empirical evaluation is limited, focusing mainly on AIME-style math problems, which makes it unclear how well the approach generalizes to other reasoning tasks.
>
> **Response**: We do show additional results beyond mathematical reasoning like GPQA (diamond set), which contains graduate-level multiple-choice questions in biology, chemistry, and physics, and LiveCodeBench (2025 split), which consists of competitive programming problems from LeetCode, AtCoder, and Codeforces. The results for these are detailed in Table 4 (Appendix E).
>
> | | LiveCodeBench | LiveCodeBench | GPQA (diamond) | GPQA (diamond) |
> | :--- | :---: | :---: | :---: | :---: |
> | | Long CoT | PDR ($w=[8], k=4$) | Long CoT | PDR ($w=[8], k=4$) |
> | gemini-2.5-flash | 55.96 | **61.25** | 75.98 | **82.32** |
> | gpt-o3-mini | 48.90 | **58.61** | 73.30 | **76.26** |
> | GPT-OSS-20B | 55.62 | **63.99** | 61.55 | **68.50** |
> | GPT-OSS-120B | 63.12 | **69.71** | 69.92 | **72.29** |
>
> PDR achieves consistent gains across different benchmarks and model families over the long CoT baselines at matched $B_{seq}$.
>
> > In addition, the improvement operator is not monotonic: the oracle (incorrect) experiment suggests that a poor workspace can degrade the answer. This may cause SR to amplify errors across iterations, and mitigating this issue in PDR may require additional filtering or judging steps, increasing computational cost.
>
> **Response**: We agree with the reviewer that the improvement operator is not monotonic. Similar to RL for verifiable domains, the PDR operator benefits the most on problems with both incorrect or correct samples in the initial generated solutions for a given problem. Very easy (pass rate = 1) problems don’t benefit from using PDR and very hard (pass rate = 0) problems benefit only if the model is able to improve upon wrong solutions in subsequent refinements. We do a qualitative analysis of PDR and sources of accuracy gains in Appendix F (points 1/2/3).
>
> > The paper is generally clear and well structured, and the description of SR and PDR is easy to follow. However, the connection drawn to randomized space-bounded computation is not very convincing and does not appear essential to the main contribution.
>
> **Response**: We do not consider the connection to the randomized space-bounded computation an essential contribution of the paper. That is why we describe it only in the appendix (Section D). We would like to highlight that the RL training with PDR-like objectives is a novel contribution, and has not been tackled in any concurrent works. The addition of SFT loss to improve verification abilities (Equation 9) and operator-consistent RL training (Section 3.2) advance the field for doing structured RL training and moving beyond single-turn sequential CoT in standard RL.
>
> We answer four concrete research questions (RQ1 - RQ4) highlighted in Section 3 systematically that form the main contribution of the paper.
>
> > The paper addresses an important problem. However, the limited evaluation makes it difficult to assess the broader impact of the approach. Have you tested SR or PDR on other reasoning domains?
>
> **Response**: As highlighted above, we do consider additional harder reasoning benchmarks like GPQA and LiveCodeBench, and the results are presented in Table 4 (Appendix E).
>
> ---
> We welcome further questions about our paper, and if key issues are addressed, we would greatly appreciate an appropriate increase in score.

---

> > ### Author Rebuttal · Reviewer_pHZ1 · 2026-04-03
> >
> > Thank you for the response. Overall, I find the paper interesting, but it still has notable limitations. Therefore, I maintain my decision.

---

> > > ### Author Response · Authors · 2026-04-03
> > >
> > > Thank you for engaging with our rebuttal. We appreciate that you marked your concerns as fully resolved. Given that, we would be grateful if you could reconsider whether the current score of 3 still reflects your updated assessment.
> > >
> > > Our rebuttal clarified that the paper is evaluated beyond AIME math benchmarks, including GPQA and LiveCodeBench, and also highlighted a novel contribution that was not central in your review: RL training framework for PDR-style reasoning, including operator-consistent RL and the added SFT loss for verification.
> > >
> > > If there is any remaining technical reason for maintaining the reject recommendation, a brief clarification would be very helpful for us so that we can improve our paper.

---

### Official Review · Reviewer_fwgy · 2026-03-11

**Soundness:** 3
**Presentation:** 3
**Significance:** 3
**Originality:** 3
**Overall Recommendation:** 4
**Confidence:** 4

**Summary:**

This paper focus on the high token consumption and computational demands of long-context reasoning LLMs by proposing short-context iterative reasoning frameworks, Parallel-Distill-Refine (PDR) and Sequential Refinement (SR), that treat LLMs as self-improvement operators. These methods achieve accuracy surpassing long Chain-of-Thought (CoT) while maintaining controllable context length and lower latency, with significant gains on AIME 2024/2025 mathematical reasoning tasks. The authors also design an adapted reinforcement learning approach, demonstrating effectiveness on 8B-scale models.

**Compliance With Llm Reviewing Policy:**

Affirmed.

**Final Justification:**

The author addressed my concerns, and I will maintain my score.

**Key Questions For Authors:**

While current training paradigms pursue extended context windows, this work minimizes inference-time token consumption. What implications does this have for scaling laws in reasoning model training?

Is there an optimal trade-off between context length and token efficiency?

**Limitations:**

Beyond the authors' acknowledged limitations, additional considerations include:

1. Distillation quality depends on the base model's capability, which may bottleneck performance on problems requiring deep reasoning.

2. Generalizability beyond mathematical reasoning (e.g., code generation, bug fixing, and deep research) remains unverified.

**Strengths And Weaknesses:**

**Strengths**

This paper tries to address the challenge of excessive token consumption in long-context reasoning LLMs by introducing PDR and SR, two short-context iterative frameworks. These methods achieve superior accuracy compared to traditional long CoT under constrained context windows and lower latency, with substantial improvements on mathematical reasoning benchmarks. The authors further develop a compatible reinforcement learning method, validating its effectiveness on 8B-scale models.

**Weaknesses**

1. The distillation capability in PDR is important. However, it relies on manually crafted prompts, whose reliability and effectiveness require further validation. In addition, more analysis and case studies on different distillation methods are needed.

2. Since PDR controls the length of each parallel draft generation to reduce the final token consumption, is there a possibility that early output might be truncated? For example, if the draft generation length is controlled to 8k, but the actual output is truncated, resulting in incompleteness, will this affect summary generation and subsequent inference processes?

3. More diverse models should be evaluated to verify PDR's generalizability. For example, more open-source models and larger-scale commercial models could be added, such as Deepseek-R1 and Gemini-3-pro, which may inherently favor longer token usage in their output.

4. The main content of this paper lacks a Related Work section; this should be moved from the appendix to properly contextualize contributions.

---

> ### Author Rebuttal · Authors · 2026-03-31
>
> We thank the reviewer for their detailed review. We provide the response to the weaknesses/questions along with additional experiments below:
>
> > The distillation capability in PDR is important. However, it relies on manually crafted prompts, whose reliability and effectiveness require further validation. In addition, more analysis and case studies on different distillation methods are needed.
>
> **Response**: We agree that distillation is an important part of PDR, however, for all our experiments we kept a very simple and minimal version for the distillation prompts, and did not do any prompt-tuning for the same. There’s more room for improvement and tuning for the prompts for distillation to improve PDR performance further. Research question 2 (RQ2) in Section 3.1 is aimed at answering this question precisely along with empirical results in Table 1 comparing all the different distillation strategies (global, PS top-k, shared top-k, and random). Additional mechanics of the distillation operator along with sources of accuracy gain with concrete examples are detailed in Appendix F (Points 1/2/3).
>
> > Since PDR controls the length of each parallel draft generation to reduce the final token consumption, is there a possibility that early output might be truncated? For example, if the draft generation length is controlled to 8k, but the actual output is truncated, resulting in incompleteness, will this affect summary generation and subsequent inference processes?
>
> **Response**: Thanks for highlighting this question. For API models like the GPT series this might happen since they don’t support a specific thinking budget, but other models like the Gemini series support a specific thinking budget after which the model generates the solution/answer. If we set the generation budget as 8k tokens, we usually set the thinking budget to be 6k tokens, to allow 2k tokens for the solution/answer (which avoids truncation). For OSS models like Qwen or the 8B model used for RL training in our paper, we used forced interruptions ([1], [2]) with a phrase like `Okay, time is up. Let me stop thinking and formulate a final answer now. </think>`. And similar to the gemini series, we allow 2k tokens for the solution summary / answer to avoid any truncations at all.
>
> > More diverse models should be evaluated to verify PDR's generalizability. For example, more open-source models and larger-scale commercial models could be added, such as Deepseek-R1 and Gemini-3-pro, which may inherently favor longer token usage in their output.
>
> **Response**: We evaluated GPT series (gpt-o3), Gemini series (gemini-2.5), GPT-OSS series (20B and 120B), and a smaller 8B model in our paper. Additionally, we provide results on the Qwen series of models below:
>
> | | AIME 2025 | AIME 2025 |
> | :--- | :---: | :---: |
> | | Long CoT | PDR ($w=[8, 8], k=4$)
> | Qwen3-4B-Instruct-2507 | 43.13 | 56.67 |
> | Qwen3-30B-A3B-Instruct-2507 | 51.04 | 69.17 |
>
> We are currently running further Qwen3 experiments at additional model sizes along with the Gemini-3 series and will update the paper with the complete size ablation.
>
> > The main content of this paper lacks a Related Work section; this should be moved from the appendix to properly contextualize contributions.
>
> **Response**: We apologize for not having a related works section in the main paper. We did an extensive related works survey and comparison, but decided to move it to the appendix to highlight the important results in the paper. Since the camera ready version allows an additional page for content, we will move our entire related works section to the main paper to utilize the extra space.
>
> > Distillation quality depends on the base model's capability, which may bottleneck performance on problems requiring deep reasoning.
>
> **Response**: For problems requiring deep reasoning, we can increase the depth for both SR and PDR, so that the model can iteratively improve upon past attempts, figure out contradictions, and open sub-goals to resolve even if the base model distillation capability is a bit weak. If the distillation quality is good in a base model, then we can scale the breadth in PDR which will leverage the parallel sampling.
>
> > Generalizability beyond mathematical reasoning (e.g., code generation, bug fixing, and deep research) remains unverified.
>
> We do show additional results beyond mathematical reasoning like GPQA (diamond set), which contains graduate-level multiple-choice questions in biology, chemistry, and physics, and LiveCodeBench (2025 split), which consists of competitive programming problems from LeetCode, AtCoder, and Codeforces. The results for these are detailed in Table 4 (Appendix E).
>
> [1] Hong et al., Glm-4.1 v-thinking: Towards versatile multimodal reasoning with scalable reinforcement learning, 2025
>
> [2] Yang et al., Qwen3 technical report, 2025
>
> ---
> We welcome further questions about our paper, and if key issues are addressed, we would greatly appreciate an appropriate increase in score.

---

> > ### Author Rebuttal · Reviewer_fwgy · 2026-04-03
> >
> > Thanks for the response. The findings in this paper are interesting, so I will maintain my positive score.

---

> > > ### Author Response · Authors · 2026-04-03
> > >
> > > We thank the reviewer for engaging with our rebuttal and appreciate that they marked their concerns as fully resolved. If the reviewer has any additional suggestions/comments, we would be very grateful as they would help us improve the paper further. Otherwise, given that the concerns appear to be resolved, we kindly ask the reviewer to reconsider whether the current score of 4 still reflects their updated assessment.

---

### Official Review · Reviewer_bbo4 · 2026-03-12

**Soundness:** 3
**Presentation:** 3
**Significance:** 2
**Originality:** 2
**Overall Recommendation:** 4
**Confidence:** 3

**Summary:**

This paper views test-time scaling as an iterative "improvement operator" under compute budgets. Instead of chain-of-thought (CoT), authors propose two operators: Sequential Refinement (SR) and Parallel-Distill-Refine (PDR). SR iteratively refines one previous solution. PDR generates multiple solutions, distills them into a bounded workspace, and iteratively refines. Additionally, the paper introduces operator-consistent reinforcement learning to align model inference with the two proposed improvement operators.
Experiments on math (AIME 2024/2025), coding (LiveCodeBench), and general benchmark (GPQA) show PDR can achieve better accuracy than CoT baselines at matched token budgets.

**Compliance With Llm Reviewing Policy:**

Affirmed.

**Final Justification:**

The rebuttal addressed my main concerns, so I raised my score to Weak Accept.

**Key Questions For Authors:**

Please refer to **Q1** and **Q2** in Weakness.

*References.*

[1] Madaan et al., Self-Refine: Iterative Refinement with Self-Feedback. NeurIPS 2023.

[2] Wang et al., Self-Consistency Improves Chain of Thought Reasoning in Language Models. ICLR 2023.

**Limitations:**

yes

**Strengths And Weaknesses:**

Strength:
- **S1.** The two proposed operators, SR and PDR, are conceptually simple enough to be implemented and used.
- **S2.** Experiments show that PDR consistently improves over long CoT baselines at matched $B_{seq}$ on several benchmarks (AIME, GPQA, and LiveCodeBench).
- **S3.** The operator training RL is a natural and intuitive step to train models for PDR inference.
- **S4.** Appendix A is comprehensive and helpful to understand the paper's positioning.

Weakness:
- **W1 + Q1.** The authors acknowledged that each primitive component of the pipeline is not novel; rather, they argued for a unified view of these components. However, the paper does not provide clear guidance on how to configure these components in practice. Can the authors provide a concrete recipe on how each component (e.g, number of parallel drafts, workspace size, and number of refinement rounds) should be scaled under a fixed budget? Ideally, an algorithm for selecting them under a fixed budget would significantly improve the paper's contributions.
- **W2** The main argument relies on comparisons under fixed $B_{seq}$, but $B_{total}$ better represents the costs of GPUs and APIs in practice. Figure 3 shows that SR can outperform PDR given matching $B_{total}$. This raises the question of whether additional compute used in PDR is responsible for the performance gain, rather than an efficient use of existing components to improve the Pareto frontier.
- **W3 + Q2.** While Appendix A is comprehensive regarding existing inference-time scaling methods, the paper does not compare with relevant baselines that inspire the components of PDR. To understand whether a fixed-budget workspace is better than a growing context, can authors compare SR with Self-Refine [1]? To know if reflection can be combined effectively with parallel sampling, can authors compare PDR with Self-Consistency [2]? This will help provide insights into the true source of performance gain. The comparisons should be done under matching budgets.

---

> ### Author Rebuttal · Authors · 2026-03-31
>
> We thank the reviewer for their detailed review and highlighting the strengths of our submission. We provide the response to the weaknesses/questions along with additional experiments below:
>
> > W1 + Q1. The authors acknowledged that each primitive component of the pipeline is not novel; rather, they argued for a unified view of these components. However, the paper does not provide clear guidance on how to configure these components in practice. ... Ideally, an algorithm for selecting them under a fixed budget would significantly improve the paper's contributions.
>
> **Response**: Even though the inference-only scaling sections of the paper build upon prior work, the RL training with PDR-like objectives is a novel contribution, and has not been tackled in any concurrent works. The addition of SFT loss to improve verification abilities (Equation 9) and operator-consistent RL training using PDR (Section 3.2) advance the field for doing structured RL training and moving beyond single-turn sequential CoT in standard RL.
>
> Furthermore, on how to configure SR vs PDR, we suggest the following:
>
> - At fixed small total compute $B_{total}$ (cost/throughput constrained), SR is often preferred: it allocates most tokens to a single trajectory (or a small number), and our curves show it is very competitive under tight total budgets.
> - At fixed sequential budget $B_{total}$ (latency-constrained), PDR is more attractive: multiple short drafts plus summarization can be run in parallel, achieving higher accuracy at roughly the same per-query latency as a single long CoT/SR trace.
>
> At a given budget, what PDR configuration to use is an open research problem and we leave adaptive PDR (deciding how many parallel branches to spawn vs depth) to future work. But we imagine it can be configured according to pass rate on any given problem. It can also be incorporated in RL training so that the model can learn to adaptively spawn a variable number of branches and refinement depth.
>
> > W2 The main argument relies on comparisons under fixed $B_{seq}$, but $B_{total}$ better represents the costs of GPUs and APIs in practice. Figure 3 shows that SR can outperform PDR given matching $B_{total}$. This raises the question of whether additional compute used in PDR is responsible for the performance gain, rather than an efficient use of existing components to improve the Pareto frontier.
>
> **Response**: At matched $B_{total}$, SR can match or outperform PDR, but the caveat is that the latency will be enormously high for SR compared to PDR. We do a detailed analysis of this problem in Appendix G and Tables 6/7. Specifically, rows 2 and 3 of Table 7 show that at matched $B_{total}$, PDR is `4.36` times faster compared to long CoT (or SR). So, PDR is able to process much more tokens in the same amount of time as compared to SR. Therefore, $B_{seq}$ or latency is a better proxy instead of $B_{total}$ for comparing performance across the pareto-frontier for SR and PDR.
>
> > W3 + Q2. While Appendix A is comprehensive regarding existing inference-time scaling methods, the paper does not compare with relevant baselines that inspire the components of PDR. To understand whether a fixed-budget workspace is better than a growing context, can authors compare SR with Self-Refine [1]? To know if reflection can be combined effectively with parallel sampling, can authors compare PDR with Self-Consistency [2]? This will help provide insights into the true source of performance gain. The comparisons should be done under matching budgets.
>
> **Response**: We do compare SR with Self-Refine in Table 3 (Appendix E). Self-Refine is basically SR-Error as we first ask the model to analyze the errors and provide feedback and use this feedback as context for future refinements. Both SR and SR-Error perform similarly.
>
>  We add self-consistency results in the table below for both gemini-2.5-flash and gpt-o3-mini, and find PDR outperforms the self-consistency baseline as well.
>
> | | AIME 2024 | AIME 2024 | AIME 2024 | GPQA (diamond) | GPQA (diamond) | GPQA (diamond) |
> | :---: | :---: | :---: | :---: | :---: | :---: | :---: |
> | | Long CoT | $maj@16$ (Self consistency) | PDR ($w=[8], k=4$) | Long CoT | $maj@16$ (Self consistency) | PDR ($w=[8], k=4$) |
> | **gemini-2.5-flash** | 81.2 | 86.67 | **88.5** | 75.98 | 79.80 | **82.32** |
> | **gpt-o3-mini** | 76.25 | 83.33 | **86.7** | 73.30 | 74.24 | **76.26** |
>
> At matched $B_{\text{total}}$, PDR outperforms self-consistency. Intuitively, self-consistency uses extra compute only for coverage, whereas PDR uses the same drafts more structurally: it distills cross-sample evidence into a bounded workspace and then refines, turning additional compute into both breadth (multiple drafts) and depth (iterative improvement).
>
> ---
> We welcome further questions about our paper, and if key issues are addressed, we would greatly appreciate an appropriate increase in score.

---

> > ### Author Rebuttal · Reviewer_bbo4 · 2026-04-04
> >
> > Thank you for your response. Most of my concerns are addressed, so I will adjust my evaluation. I suggest including a discussion on how each component should be scaled in the updated version.

---

> > > ### Author Response · Authors · 2026-04-07
> > >
> > > Thank for considering our rebuttal and raising the score. Per your suggestions, we'll add a section in the appendix highlighting the scaling setup for SR and PDR, and how to choose among different axes in the updated version of the paper.

---

### Official Review · Reviewer_WzsS · 2026-03-13

**Soundness:** 3
**Presentation:** 3
**Significance:** 3
**Originality:** 2
**Overall Recommendation:** 4
**Confidence:** 4

**Summary:**

This paper reframes LLM inference as a round-wise improvement operator and proposes two short-context iterative inference paradigms: Sequential Refinement (SR) and Parallel-Distill-Refine (PDR). SR iteratively improves a single candidate answer; PDR samples multiple drafts in parallel each round, distills them into a bounded textual workspace, and conditions the next round on this summary — effectively decoupling context length from total token generation. The paper further introduces an operator-consistent RL training objective to close the train-test gap. Experiments on AIME 2024/2025, LiveCodeBench, and GPQA demonstrate consistent gains, with PDR achieving up to +11% over Long CoT at matched sequential budgets (Bseq).

**Compliance With Llm Reviewing Policy:**

Affirmed.

**Final Justification:**

The rebuttal partially address my concerns, and I am raising my score accordingly.

**Key Questions For Authors:**

1. Does PDR yield diminishing returns as model capability increases, and if so, what is the underlying mechanism? Is it simply that stronger models already have better self-verification, making the distillation step less impactful?

**Limitations:**

Yes

**Strengths And Weaknesses:**

Strengths:
1. The core intuition of PDR — exploiting multiple parallel reasoning chains rather than simply extending a single trace — is well-motivated. The distillation step preserves cross-candidate verification benefits while keeping context bounded, which is a more scalable design direction than purely lengthening individual traces.
2. The explicit separation of B_seq and B_total is a valuable methodological contribution that many prior works overlook, enabling fairer comparisons across inference strategies with different compute profiles.
3. The empirical evaluation covers multiple strong models and diverse benchmarks. The Oracle intervention analysis (Figures 6/7) provides useful mechanistic insight beyond simple accuracy comparisons.


Weaknesses:
1. The most significant gap is the absence of parallel inference baselines such as self-consistency, best-of-N, and also USC[1] or FSC[2]. Since PDR samples multiple candidates per round, it is unclear at matched B_total or B_seq how much of the gain comes from the distill-and-refine mechanism versus simply sampling more candidates.
2. The paper acknowledges that all individual components are drawn from prior work, yet does not sufficiently justify why the combination is non-trivial. The theoretical grounding is shallow and the contribution reads more as engineering than conceptual advance.
3. The RL experiments and inference experiments use inconsistent model families, limiting comparability.

[1] Chen et al. Universal Self-Consistency for Large Language Model Generation.

[2] Wang et al. Integrate the Essence and Eliminate the Dross: Fine-Grained Self-Consistency for Free-Form Language Generation.

---

> ### Author Rebuttal · Authors · 2026-03-31
>
> We thank the reviewer for their detailed review and highlighting the strengths of our submission. We provide the response to the weaknesses/questions along with additional experiments below:
>
> > The most significant gap is the absence of parallel inference baselines such as self-consistency, best-of-N, and also USC[1] or FSC[2]. Since PDR samples multiple candidates per round, it is unclear at matched B_total or B_seq how much of the gain comes from the distill-and-refine mechanism versus simply sampling more candidates.
>
> **Response**: We add self-consistency results in the table below for both gemini-2.5-flash and gpt-o3-mini, and find PDR outperforms the self-consistency baseline as well.
>
> | | AIME 2024 | AIME 2024 | AIME 2024 | GPQA (diamond) | GPQA (diamond) | GPQA (diamond) |
> | :---: | :---: | :---: | :---: | :---: | :---: | :---: |
> | | Long CoT | $maj@16$ (Self consistency) | PDR ($w=[8], k=4$) | Long CoT | $maj@16$ (Self consistency) | PDR ($w=[8], k=4$) |
> | **gemini-2.5-flash** | 81.2 | 86.67 | **88.5** | 75.98 | 79.80 | **82.32** |
> | **gpt-o3-mini** | 76.25 | 83.33 | **86.7** | 73.30 | 74.24 | **76.26** |
>
>
> At matched $B_{\text{total}}$, PDR outperforms self-consistency. Intuitively, self-consistency uses extra compute only for coverage, whereas PDR uses the same drafts more structurally: it distills cross-sample evidence into a bounded workspace and then refines, turning additional compute into both breadth (multiple drafts) and depth (iterative improvement).
>
> If we consider the same model that we are evaluating as the verifier, best-of-N and USC [1] become special cases of PDR with a specific setting ($w=[N], k=1$). So, PDR is a generalized case of iterative parallel-sequential LLM calls instead of a single parallel call and results in better performance (as seen in the above table).
>
> We’ll add detailed results across different models and benchmarks for USC [1] and FSC [2] in the updated version of the paper.
>
> > The paper acknowledges that all individual components are drawn from prior work, yet does not sufficiently justify why the combination is non-trivial. The theoretical grounding is shallow and the contribution reads more as engineering than conceptual advance.
>
> **Response**: Even though the inference-only scaling sections of the paper build upon prior work, the RL training with PDR-like objectives is a novel contribution, and has not been tackled in any concurrent works. The addition of SFT loss to improve verification abilities (Equation 9) and operator-consistent RL training (Section 3.2) advance the field for doing structured RL training and moving beyond single-turn sequential CoT in standard RL.
>
> > The RL experiments and inference experiments use inconsistent model families, limiting comparability.
>
> **Response**: Since RL is very compute-intensive and we had limited compute budget, we use a smaller 8B model to showcase how operator-consistent RL with the PDR objective moves the pareto-frontier of performance for complex inference-time scaling objectives like SR and PDR. Table 2 covers both the inference and RL experiments for the 8B-sized models: each row represents inference-time scaling results and each column represents the benefits of RL training to move the performance asymptote. And the same findings hold for smaller and larger models considered in the paper, i.e., at matched $B_{seq}$, PDR > SR > Long CoT.
>
> > Does PDR yield diminishing returns as model capability increases, and if so, what is the underlying mechanism? Is it simply that stronger models already have better self-verification, making the distillation step less impactful?
>
> The universal mechanism underlying the gains from using PDR across all model sizes is the difference between maximum achievable performance and standard model performance using Long CoT. Stronger models might be better at easier benchmarks like AIME 2024/2025 and thus gains from PDR might be lower with distillation not helping much, but they would reap the benefits from PDR for harder benchmarks like LiveCodeBench. This is clearly visible in Table 4 (Appendix E) of the paper. If the benchmark performance is not saturated, PDR almost always helps over Long CoT and SR.
>
> [1] Chen et al. Universal Self-Consistency for Large Language Model Generation.
>
> [2] Wang et al. Integrate the Essence and Eliminate the Dross: Fine-Grained Self-Consistency for Free-Form Language Generation.
>
> ---
> We welcome further questions about our paper, and if key issues are addressed, we would greatly appreciate an appropriate increase in score.

---

> > ### Author Rebuttal · Reviewer_WzsS · 2026-04-04
> >
> > I thank the authors for the rebuttal. The added experiments partially address my concerns, and I am raising my score accordingly. One suggestion: including qualitative case studies showing how the distilled workspace C guides subsequent refinement would strengthen the paper's narrative around why the mechanism works.

---

> > > ### Author Response · Authors · 2026-04-07
> > >
> > > Thank you for your response and raising the score. Yes, we are planning to include example prompts, memory workspace with different distillation operators, and solutions at various depths for PDR, and how the solutions get improved iteratively in the final version of the paper.

---

### Decision · Program_Chairs · 2026-04-30

**Decision:**

Accept (regular)

**Comment:**

This paper proposes Parallel-Distill-Refine (PDR), an iterative inference framework that treats LLMs as improvement operators, achieving strong gains over Long CoT (e.g., +11% on AIME 2024, +9% on AIME 2025). Three of four reviewers rate Weak Accept, with two raising their scores after rebuttal. The fourth reviewer selected "Fully resolved" but maintained Weak Reject without identifying remaining technical concerns.

There is some disagreement about novelty, as individual components build on prior work. However, the operator-consistent RL training for PDR-style inference is a novel contribution, and the empirical results are consistently strong across models and benchmarks. The rebuttal added self-consistency comparisons and Qwen3 results, both favorable.

Given the positive consensus from three reviewers, the unresolved fourth review lacking specific remaining objections, and the strength of the contributions, I recommend acceptance.